# SSG-ECPE: Semantics-Structured Generation with Alignment for Emotion-Cause Pair Extraction

## Abstract

Emotion Cause Pair Extraction (ECPE) aims to jointly identify emotion clauses and their corresponding cause clauses, forming emotion-cause pairs (ECPs). Existing approaches either rely on complex discriminative architectures to model pair boundaries or adopt generic text-to-text frameworks that flatten ECPE into plain sequence generation. Both paradigms overlook rich semantic dependencies, such as clause roles, emotion types, and clue words, and struggle in multi-pair scenarios with nested or overlapping structures. In this paper, we propose a task-adaptive generative multi-task learning framework that rethinks ECPE as a structured text-to-text generation task. We design semantics-structured output formats that explicitly encode clause roles, emotion types, and trigger words as semantic markers, allowing the model to capture inter-label dependencies and co-occurrence patterns during generation. For emotion clause extraction (EE), outputs are formatted as *(clause, emotion type, trigger words)* triplets; for ECPE, emotion–cause pairs are directly generated, enabling implicit modeling of emotional reasoning. A shared encoder with task-specific decoders supports both clause- and pair-level generation within a unified pipeline. To enhance reliability, we further introduce a Clause Prediction Alignment (CPA) strategy that grounds generated clauses to input spans, mitigating hallucinations and ensuring faithfulness. Extensive experiments demonstrate that CPA is indispensable: without it, performance collapses, whereas with it, our framework achieves consistent state-of-the-art results, including a +21.3 F1 improvement on the English benchmark.

## 1 Introduction

Sentiment analysis (SA) is a fundamental topic in artificial intelligence and natural language processing, aiming to identify and understand the emotions or opinions expressed in texts. Traditional SA studies Li et al. (2015); Hu et al. (2022) focus on coarse-grained emotion categories, such as sentiment polarity (positive, neutral, negative). However, such shallow classification often fails to meet the demands of more complex analytical needs. As research progresses, increasing attention has shifted toward Emotion Cause Analysis (ECA) Chen et al. (2020a); Weng et al. (2020), which not only detects emotions but also identifies their underlying causes. This task is crucial for psychological research and has broad applications in sociology, marketing, and education. For example, it aids in creating effective treatment plans in psychotherapy, helps companies understand consumer behavior in marketing, and supports personalized instruction in education Weng et al. (2020).

Xia & Ding (2019) proposed the ECPE task, which simultaneously extracts all emotions and corresponding causes from the unannotated text. Specifically, the objective of the ECPE task is to extract all the emotion and cause clauses from a given document at once and form ECPs. Figure 1 shows an example with 8 clauses. $c_1$ and $c_8$ express the emotions "disgust" and "surprise", respectively. $c_1$ has two causes: $c_2$ and $c_3$, while $c_8$ is triggered by $c_5$ and $c_7$. The desired ECPs are $\{(c_1, c_2), (c_1, c_3), (c_8, c_5), (c_8, c_7)\}$. This example highlights the challenges of ECPE, including multiple emotions, overlapping causes, and nested structures.

Existing ECPE methods can be categorized into 2-step pipeline Xia & Ding (2019), multi-task learning Chen et al. (2022b); Zheng et al. (2022); Shang et al. (2023); Fu & Li (2024), sequence

labeling Fan et al. (2020); Cheng et al. (2021), graph-based Bao et al. (2022); Li et al. (2023b); Zhu et al. (2024); Li et al. (2024), question answering (QA) Nguyen & Nguyen (2023), machine reading comprehension (MRC) Zhou et al. (2022); Cheng et al. (2023); Mai et al. (2024), reinforcement learning (RL) framework Chen et al. (2023). Despite their diversity, most adopt a discriminative paradigm: they first generate candidate clause pairs and then classify them as valid or invalid through complex feature engineering or architectures.

---

**Input: A Document**

C1: Life is better than death is what Wu often thinks after suffering from cancer
(生不如死是老吴患癌后常有的想法) **_emo: disgust**
C2: Due to extensive bone metastases (由于大面积骨转移) **_cau**
C3: Wu struggled with severe pain every day (老吴每天都在剧痛里挣扎) **_cau**
C4: From October 2012 to now (从2012年10月到现在)
C5: Wu has been battling advanced lung cancer for 30 months (老吴已经和晚期肺癌搏斗了30个月) **_cau**
C6: this length of time (这个时间长度)
C7: this length of time is nearly 10 times longer than the initial death sentence given to him by the doctors
(比医生最初给他的死亡判决已经超出了近10倍) **_cau**
C8: Doctor Wang Jin of the Shenzhen Hospice was also surprised
(深圳宁养院的医生王劲也感到惊诧) **_emo: surprise**

**Output: ECPs (Emotion_clause:content, Cause_clause:content)**

Emotion clause:C1: Life is better than death is what Wu often thinks after suffering from cancer, Cause clause:C2: Due to extensive bone metastases, C3: Wu struggled with severe pain every day;
Emotion clause:C8: Doctor Wang Jin of the Shenzhen Hospice was also surprised, Cause clause:C5: Wu has been battling advanced lung cancer for 30 months, C7: this length of time is nearly 10 times longer than the initial death sentence given to him by the doctors

Figure 1: An example of an ECPE task based on generative framework from the Chinese dataset. Emotion clauses are shown in red, and cause clauses are shown in blue. Various shades of color distinguish different emotions and cause clauses.

However, these approaches still face several limitations. (1) they rely heavily on high-quality annotations and are sensitive to distributional shifts. (2) they lack a global view of clause-level semantics and inter-clausal dependencies, resulting in poor performance in cross-clause reasoning and difficulty in dealing with complex multiple ECPs. (3) these models ignore semantic labels, such as the emotion types, clause roles, and emotion trigger words. Encoding such semantic knowledge can significantly improve the modeling of documents with multiple ECPs. For example, as shown in Figure 1, recognizing that "disgust" is semantically associated with severe pathological conditions like "extensive bone metastasis" and enduring physical suffering such as "struggled with severe pain every day" allows the model to link $c_1$ to both $c_2$ and $c_3$, leading to more accurate extraction of the set of ECPs $\{(c_1, c_2), (c_1, c_3)\}$.

Recent advances in generative models have shown strong performance in structured NLP tasks by reformulating them as text-to-text generation problems Lu et al. (2022); Wang et al. (2023a). Unlike discriminative models that make isolated decisions over candidate pairs, generative models naturally model inter-label dependencies through autoregressive decoding, making them well-suited for tasks with complex, structured outputs.

Inspired by this, we propose to rethink ECPE as a semantics-structured generation task. By explicitly encoding label semantics, such as emotion types (i.e., happiness), clause roles (i.e., emotion/cause clause), and emotion trigger words (i.e., surface expressions that signal or instantiate emotional states, such as: "surprised at"), into the output format, our model can leverage the meaning of labels to guide generation. For instance, the presence of the emotion "disgust" can prompt the model to seek clinically negative events as potential causes. This enables implicit modeling of emotional reasoning, going beyond mere pattern matching.

Our main contributions are summarized as follows:

- We reformulate ECPE as a structured text-to-text generation task, integrating clause roles, emotion types, and emotion trigger words into the output format to enable explicit modeling of semantic dependencies between emotions and causes.

- We design a multi-task generative framework (SSG-ECPE) with a shared encoder and task-specific decoders, allowing joint training of EE and ECPE within a unified architecture to enhance cross-task knowledge transfer.

- We introduce a clause prediction alignment strategy that constrains generated clauses to match actual clauses in the input document, effectively reducing hallucinations and improving faithfulness.

- Extensive experiments on benchmark datasets show that SSG-ECPE achieves state-of-the-art performance, with significant gains over existing methods (e.g., +21.31 F1 on the English ECPE dataset).

## 2 RELATED WORK

**Discriminative Models for ECPE.** Most existing ECPE methods follow a discriminative paradigm, aiming to identify valid ECPs by classifying candidate clause pairs using sophisticated feature engineering or interaction modeling. Early approaches adopted pipeline frameworks Xia & Ding (2019), where emotion and cause clauses are extracted separately and then paired. However, this paradigm suffers from error propagation and fails to model inter-task dependencies.

To address these issues, joint learning frameworks were proposed to unify emotion clause extraction (EE), cause clause extraction (CE), and ECPE format within a single model Chen et al. (2022b); Shang et al. (2023); Li et al. (2023a). While enabling task interaction, these methods often face task imbalance and insufficient semantic alignment. Alternative views regard ECPE as a sequence labeling problem using hand-crafted tagging schemes (e.g., BIOE or pair-level tags) Fan et al. (2021); Cheng et al. (2021). However, such approaches struggle with generalization due to heuristic design. Graph-based methods enhance clause interaction modeling by representing documents as structured graphs. For instance, Fan et al. (2020) converts the ECPE task to a directed graph construction, while Chen et al. (2020b); Liu et al. (2022) explicitly model ECP relations. However, they often underperform in long-range scenarios unless augmented with external knowledge (e.g., commonsense, clause dependencies) Bao et al. (2022); Li et al. (2023b). Multi-granularity models Chen & Mao (2023) integrate word-, clause-, and document-level semantics but still struggle with complex relational structures. Other works explore question answering (QA) or machine reading comprehension (MRC) paradigms, treating ECPE as a query-based extraction task Zhou et al. (2022); Cheng et al. (2023); Mai et al. (2024); Nguyen & Nguyen (2023). Despite competitive performance, these methods are constrained by predefined templates and exhibit limited effectiveness on long documents.

**Generative Models for ECPE.** Inspired by the success of structured text-to-text generation in information extraction Lu et al. (2022); Wu et al. (2022); Wang et al. (2023a), recent studies have explored generative formulations for ECPE, which naturally model inter-label dependencies through autoregressive decoding. Zheng et al. (2022) propose a multi-task prompt framework that decomposes ECA tasks into sub-prompts for unified modeling. However, this approach relies on manually designed templates, limiting its flexibility and generalization. With the rise of large language models (LLMs), zero-shot and few-shot ECPE has gained attention. Wang et al. (2023b) apply ChatGPT to ECPE, leveraging its semantic understanding, but suffer from uncontrolled outputs and weak task-specific adaptation. DECC Wu et al. (2024) introduces a chain-of-thought strategy to decompose ECPE into sub-tasks, yet faces high computational costs and suboptimal performance in complex, multi-pair scenarios.

However, they still face some issues: (1) hallucination: generating non-existent clauses, (2) failing to fully leverage label semantics (e.g., emotion types, clause roles). Our work addresses these limitations by introducing a semantics-structured output format and a clause-level alignment mechanism, ensuring more reliable and coherent predictions.

## 3 METHOD

We propose Semantics-Structured Generation with Alignment (SSG-ECPE), a generative framework for Emotion-Cause Pair Extraction (ECPE). As illustrated in Figure 2, our method reformulates ECPE and its auxiliary task, Emotion Clause Extraction (EE), into conditional sequence generation problems. Specifically, SSG-ECPE is built upon a shared encoder and two task-specific decoders,

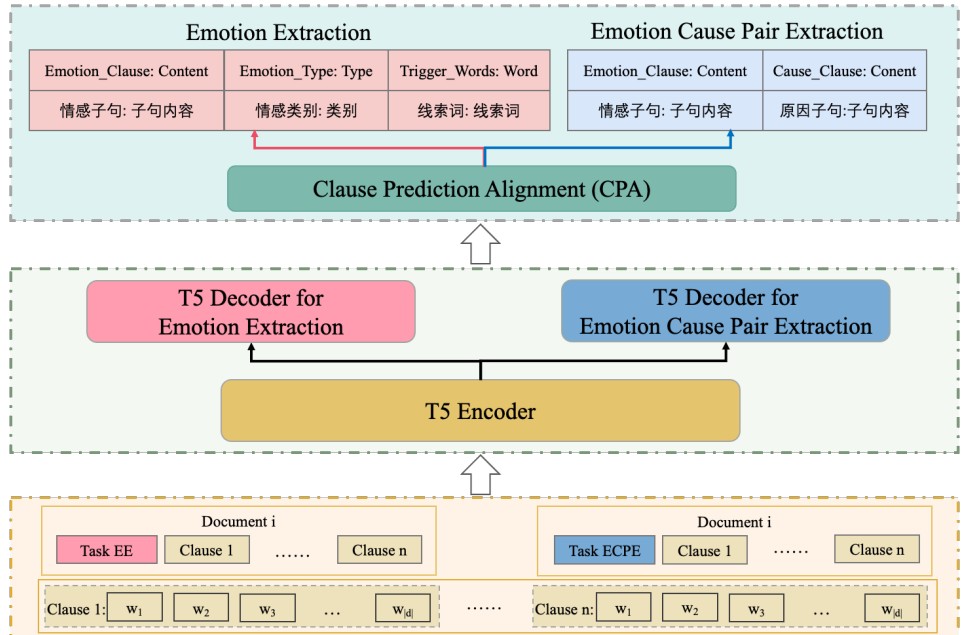

Figure 2: An example of an ECPE task based on generative framework from the Chinese dataset.

allowing knowledge sharing across tasks while maintaining task-level specialization. To ensure the generated outputs remain faithful to the input document, we introduce two key components: (1) a semantics-structured generation format, which explicitly incorporates clause roles, emotion types, and trigger words into the target sequences; and (2) a clause prediction alignment (CPA) strategy, which aligns generated spans with the closest input clauses to mitigate hallucinations and guarantee clause-level consistency. This design enables SSG-ECPE to handle complex phenomena in ECPE, such as multiple causes for one emotion, overlapping relations, and self-referential ECPs, while producing interpretable and semantically grounded outputs.

## 3.1 PROBLEM FORMULATION

Given a document with $n$ clauses $D = \{c_1, c_2, \ldots, c_n\}$, each clause contains multiple words $c_i = \{w_{i,1}, w_{i,2}, \ldots, w_{i,|c_i|}\}$. The ECPE task aims to extract all valid ECPs $(c_i, c_j)$, where clause $c_i$ expresses an emotion and $c_j$ provides its cause:

$$y_{ECPE} = \{(c_i, c_j)\}_{(i,j) \in \mathcal{P}} \tag{1}$$

where $\mathcal{P}$ denotes the set of valid clause index pairs.

Unlike traditional formulations that treat ECPE as a classification or matching problem, we reformulate it under a structured generation paradigm. Specifically, the task is cast as a constrained text generation problem: given the input document $D$, the model generates a structured sequence that encodes all valid ECPs in a predefined format. This formulation not only accommodates many-to-many relationships but also naturally handles cases where an emotion clause also functions as its own cause clause.

## 3.2 MULTI-TASK GENERATIVE FRAMEWORK

**Shared Encoder**. To jointly capture the semantic dependencies across emotion extraction (EE) and emotion-cause pair extraction (ECPE), we adopt a multi-task generative framework built on the T5 architecture. A shared encoder is employed to encode the entire document with explicit clause boundary markers. The input sequence is constructed as:

$$X = \texttt{[TASK]} \oplus c_1 \oplus \texttt{<c>} \oplus c_2 \oplus \texttt{<c>} \oplus \cdots \oplus \texttt{<c>} \oplus c_n \oplus \texttt{<c>} \tag{2}$$

| Input: |
|---|
| Task Descriptions:c1:The family currently relies on her for more than 2,000 yuan in wages to live, c2:the medical treatment has been heavily indebted, c3:if I disappeared, c4:and my wife did not have a Shenzhen household registration, c5:the days would be even more difficult, c6:it is helpless and heartbreaking, c7:but I also appreciate the netizens who help me, c8:and the family's unwavering commitment, c9:now I can only grit my teeth and continue to hold on, c10:and I hope to achieve the wish. |
| **Ground Truth**: |
| Emotion Clauses: c7:but I also appreciate the netizens who help me |
| Cause Clauses: c7:but I also appreciate the netizens who help me |
| c8:and the family's unwavering commitment |
| Emotion Types: happiness |
| ECPs: $\{c7, c7\}, \{c7, c8\}$ |
| **Target**: |
| **EE Target**: [emotion clause: c7:but I also appreciate the netizens who help me, emotion type: happiness, keywords: appreciate] |
| **ECPE Target**: [emotion clause: c7:but I also appreciate the netizens who help me, cause clause: c7:but I also appreciate the netizens who help me, c8:and the family's unwavering commitment] |

Table 1: Example of EE and ECPE tasks. The input is task description and all clauses. The target is the ECPs or emotion clauses.

where $\oplus$ denotes concatenation, and $\texttt{<c>}$ is a special token marking clause boundaries. $\texttt{[TASK]}$ is a task-specific prefix (e.g., "EE:" or "ECPE:"). The shared encoder $\mathcal{E}_\theta^{\text{shared}}$ produces contextualized token-level representations:

$$H = \mathcal{E}_\theta^{\text{shared}}(X) \in \mathbb{R}^{L \times d} \tag{3}$$

where $L$ is the sequence length, and $d$ is the hidden dimension.

**Task-Specific Decoders**. On top of the shared encoder, we employ task-specific decoders to generate structured outputs for different tasks. Specifically, two separate decoders, $\mathcal{D}_\phi^{\text{EE}}$ and $\mathcal{D}_\psi^{\text{ECPE}}$, are responsible for EE and ECPE respectively. Each decoder conditions on the shared representation $H$ to produce task-specific sequences:

$$Y^{\text{EE}} = \mathcal{D}_\phi^{\text{EE}}(H), \quad Y^{\text{ECPE}} = \mathcal{D}_\psi^{\text{ECPE}}(H) \tag{4}$$

### 3.3 Semantics-Structured Generation Format

To unify EE and ECPE within a single generative framework, we design a *semantics-structured generation format* that explicitly incorporates rich label semantics into the model outputs. These semantics include emotion types, clause roles (i.e., *emotion* vs. *cause* clauses), and emotion trigger words, allowing the model to produce outputs that are both structured and interpretable. Our framework naturally supports self-referential ECPs $(c_i, c_i)$, which occur when a single clause simultaneously expresses an emotion and its underlying cause, as in self-reflective statements (e.g., "I am happy because I am grateful").

**EE Generative Paradigm**. For the EE task, the goal is to identify all clauses expressing emotions and determine their corresponding emotion type and trigger words. We define the generative target as:

$$G_{EE} = [\underline{\text{emotion clause}} : c_i, \underline{\text{emotion type}} : \text{emo}, \underline{\text{keywords}} : kw] \tag{5}$$

where $c_i \in D$ is an emotion clause, $emo \in \varepsilon = \{$anger, disgust, fear, happy, sad, surprise$\}$, $kw$ is a salient token or phrase indicating the emotional expression. This structured output explicitly encodes emotion semantics and facilitates modeling co-occurrence patterns between emotion types and their lexical triggers. Table 1 is an illustrative example of EE tasks.

**ECPE Generative Paradigm**. The ECPE task aims to identify all valid ECPs in a document. The generation target is defined as:

$$G_{ECPE} = [\underline{\text{emotion clause}} : c_i, \underline{\text{cause clause}} : c_j] \tag{6}$$

When an emotion clause $c_i$ is associated with multiple causes $\{c_{j_1}, c_{j_2}, ..., c_{j_k}\}$, we aggregate them in a single structured entry:

$$[\text{emotion clause} : c_i, \text{cause clause} : c_{j_1}, c_{j_2}, ..., c_{j_k}] \tag{7}$$

which preserves the natural clustering of causal information. As shown in Table 1, this format supports scenarios where the emotion clause is also its own cause (e.g., clause c7) while accommodating additional external causes.

## 3.4 OUTPUT DECODING STRATEGY

Given an input document, the generative model produces a target sequence $Y'$ following the pre-defined semantics-structured schema. Each output entry is enclosed within square brackets `[]` to provide explicit boundaries, which facilitates automated parsing and reduces ambiguity in the generated sequences.

For the ECPE task, we adopt an *emotion-centric causal clustering* strategy. Concretely, for each unique emotion clause $c_i$ that corresponds to one or more cause clauses, the model generates a single structured entry of the form: `[emotion clause: <content>, cause clause: <cause_1>, <cause_2>, ..., <cause_k>]`. This design naturally supports: (1) multiple causes for a single emotion clause, and (2) self-referential pairs where the emotion clause itself also serves as its own cause $(c_i, c_i)$. All entries in a document are concatenated and separated by semicolons `;`, as exemplified in Table 1.

Similarly, for the EE task, the output consists of one or more entries formatted as: `[emotion clause: <content>, emotion type: <type>, keywords: <kw>]`. Multiple entries are also separated by semicolons `;`.

The parsing process is as follows: (1) Split the generated sequence by `;` to extract individual emotion-centric entries. (2) For each entry, extract the emotion clause content. (3) For ECPE, split the content following `cause clause:` by `,` to obtain all associated cause clauses. (4) Form ECPs by pairing the emotion clause with each listed cause clause.

To address potential generation errors or malformed outputs, we implement a *recovery strategy* that identifies the longest well-formed substring within brackets. Only valid entries are parsed, while malformed segments are ignored, ensuring that extraction remains robust and faithful to the original document. This mechanism guarantees the reliability of the final ECPs even under imperfect generation conditions, accommodating multi-cause scenarios and self-referential ECPs.

## 3.5 CLAUSE PREDICTION ALIGNMENT

Generative models occasionally produce clauses with minor lexical variations or paraphrasing, which may deviate from the original document clauses. To enforce faithful clause-level predictions, we introduce **Clause Prediction Alignment (CPA)** as a simple yet effective post-processing step. Formally, for each generated clause $c$, we search for the most similar clause $c^* \in D$ within the original document by maximizing a normalized sequence similarity:

$$c^* = \arg\max_{c' \in D} \text{Sim}(c, c'), \tag{8}$$

where $\text{Sim}(c, c') \in [0, 1]$ is computed using the `SequenceMatcher` algorithm (based on longest common subsequence matching). If the highest similarity exceeds a threshold $\tau$, we replace $c$ with $c^*$; otherwise, we retain the original prediction. This thresholding mechanism prevents overcorrection and ensures only reliable alignments are applied.

Compared to exact edit-distance matching, the LCS-based similarity measure is more tolerant to minor rephrasings while preserving sequence-level fidelity. This ensures that all final predictions are grounded in the predefined clause set $D$, with a threshold $\tau$ tuned on the validation set to avoid overcorrection. Please refer to Appendix B for the detailed implementation.

## 3.6 MULTI-TASK LEARNING OBJECTIVE

Our framework jointly models EE and ECPE in a multi-task generative setting. To this end, we employ a shared encoder $\mathcal{E}_\theta$ to capture common contextual representations, while task-specific decoders $\mathcal{D}_\phi^{\text{EE}}$ and $\mathcal{D}_\psi^{\text{ECPE}}$ generate outputs for their respective tasks. This design enables knowledge transfer between EE and ECPE: emotion semantics captured in EE can guide ECPE in identifying causal relationships, and vice versa.

For the EE task, given the input $x_{EE} = \text{EE:} \oplus D$ and encoder output $H = \mathcal{E}_\theta(x_{EE})$, the training objective maximizes the likelihood of the target token sequence:

$$\mathcal{L}_{EE}(x) = -\frac{1}{N_x} \sum_{i=1}^{N_x} \log P_\phi\big(y_i^{\text{EE}} \mid H_x, y_{<i}^{\text{EE}}\big) \tag{9}$$

where $y_{<i}^{\text{EE}} = \{y_1^{\text{EE}}, \ldots, y_{i-1}^{\text{EE}}\}$ denotes the tokens generated before step $i$, and $N$ is the length of the EE target sequence.

Similarly, for the ECPE task with input $x_{ECPE} = \texttt{ECPE:} \oplus D$, the negative log-likelihood loss is:

$$\mathcal{L}_{ECPE} = -\frac{1}{M} \sum_{j=1}^{M} \log P_\psi \left( y_j^{\text{ECPE}} \mid H, y_{<j}^{\text{ECPE}} \right) \tag{10}$$

where $M$ is the length of the ECPE target sequence. The overall training objective is a weighted combination of the two losses:

$$\mathcal{L} = \lambda_{EE} \cdot \mathcal{L}_{EE} + \lambda_{ECPE} \cdot \mathcal{L}_{ECPE} \tag{11}$$

where $\lambda_{EE}$ and $\lambda_{ECPE}$ are task-balancing hyperparameters, tuned on the validation set.

By sharing the encoder, our model leverages cross-task information, allowing EE to inform the model about salient emotional content, while ECPE benefits from capturing causal dependencies among clauses. At the same time, the task-specific decoders retain flexibility to specialize in their respective outputs.

## 4 Experiments

### 4.1 Experimental Settings

**Dataset.** We evaluate our **SSG-ECPE** framework on two widely-used benchmark datasets: (1) *Chinese ECPE Dataset* Xia & Ding (2019): A clause-level annotated Chinese corpus specifically designed for ECPE. (2) *NTCIR-13 English Emotion Corpus* Gao et al. (2017): An English dataset extracted from novels. Comprehensive dataset statistics are provided in the Appendix D.

**Evaluation Metrics.** For the evaluation metrics, similarly to the prior work Xia & Ding (2019), we report the Precision, Recall, and F1-score for the main ECPE task, as well as for the auxiliary sub-tasks of emotion clause extraction (EE) and cause clause extraction (CE).

**Baselines** To comprehensively evaluate the effectiveness of our proposed **SSG-ECPE** framework, we compare it against a wide range of representative baselines on both Chinese and English datasets. Unless otherwise specified, all supervised fine-tuning (SFT) baselines are implemented with BERT-based encoders to ensure fair comparison. We group the baselines into the following categories: **(1) Discriminative SFT Models.** These methods formulate ECPE as a discriminative clause-pair classification or tagging task. Representative works include Indep/Inter-CE/Inter-EC Xia & Ding (2019), RANKCP Wei et al. (2020), ECPE-2D Ding et al. (2020a), PairGCN Chen et al. (2020b), ECPE-MLL Ding et al. (2020b), Tagging Yuan et al. (2020), and subsequent refinements such as UTOS Cheng et al. (2021), Refinement Fan et al. (2021), and CD-MRC Cheng et al. (2023). **(2) Graph- and Multi-task-based Models.** These approaches exploit syntactic/semantic graphs or auxiliary tasks to enhance representation learning. Examples include PairGCN Chen et al. (2020b), KMGP Zong et al. (2024), RSN Chen et al. (2022a), MGSAG Bao et al. (2022), A$^2$Net Chen et al. (2022b), ECPE-MTL Li et al. (2023a), and MMN Shang et al. (2023). **(3) Generative and Prompt-based Models.** Recent work explores generative paradigms or prompting strategies, including UECA-Prompt Zheng et al. (2022), DECC Wu et al. (2024), and LLM-based approaches such as GPT-3.5 Wang et al. (2023b) and DeepSeek-V3 DeepSeek-AI et al. (2024). **(4) English-specific Baselines.** For the NTCIR-13 English dataset, in addition to the above, we include E2E-PExtE Singh et al. (2021) and IA-ECPE Huang et al. (2023), which are tailored to English corpora.

**Implementation Details** All experiments are conducted under the T5 framework. Specifically, we adopt `Randeng-T5-77M-MultiTask-Chinese` for the Chinese dataset and `T5-base` for the English dataset. For the Chinese dataset, we consider two widely-used experimental setups for fair and comprehensive comparison: **Setting 1**: Following Xia & Ding (2019), the dataset is split into 90% training and 10% testing. **Setting 2**: Following Fan et al. (2020), the dataset is divided into 80%/10%/10% for training/validation/testing, where the model is fine-tuned on the validation set and evaluated on the test set. For the English dataset, we followed Singh et al. (2021) with an 80%/10%/10% split. In both datasets, we employ AdamW as the optimizer with a learning rate of $3 \times 10^{-4}$, batch size of 24, and train for 20 epochs. Detailed experimental configurations can be found in the Appendix C.

| | Main Task | | | Auxiliary Task | | | | | |
| Approach | Emotion-Cause Pair Extraction | | | Emotion Clause Extraction | | | Cause Clause Extraction | | |
| | P | R | F1 | P | R | F1 | P | R | F1 |
| --- | --- | --- | --- | --- | --- | --- | --- | --- | --- |
| **Setting 1: 90% for training, 10% for testing** | | | | | | | | | |
| Indep | 0.6832 | 0.5082 | 0.5818 | 0.8375 | 0.8071 | 0.8210 | 0.6902 | 0.5673 | 0.6205 |
| Inter-CE | 0.6902 | 0.5153 | 0.5901 | 0.8494 | 0.8122 | 0.8300 | 0.6809 | 0.5634 | 0.6151 |
| Inter-EC | 0.6721 | 0.5705 | 0.6128 | 0.8364 | 0.8107 | 0.8230 | 0.7041 | 0.6083 | 0.6507 |
| EDSECPE† | 0.7822 | 0.7417 | 0.7614 | 0.9243 | 0.9115 | 0.9179 | 0.7981 | 0.7821 | 0.7900 |
| RANKCP† | 0.7119 | 0.7630 | 0.7360 | 0.9123 | 0.8999 | 0.9057 | 0.7461 | 0.7788 | 0.7615 |
| ECPE-2D† | 0.7292 | 0.6544 | 0.6889 | 0.8627 | 0.9221 | 0.8910 | 0.7336 | 0.6934 | 0.7123 |
| PairGCN† | 0.7692 | 0.6791 | 0.7202 | 0.8857 | 0.7958 | 0.8375 | 0.7907 | 0.6928 | 0.7375 |
| ECPE-MLL† | 0.7700 | 0.7235 | 0.7452 | 0.8608 | 0.9191 | 0.8886 | 0.7382 | 0.7912 | 0.7630 |
| RSN† | 0.7601 | 0.7219 | 0.7393 | 0.8614 | 0.8922 | 0.8755 | 0.7727 | 0.7398 | 0.7545 |
| MGSAG† | 0.7743 | 0.7321 | 0.7521 | 0.9208 | 0.8211 | 0.8717 | 0.7979 | 0.7468 | 0.7712 |
| A$^2$Net† | 0.7503 | 0.7780 | 0.7634 | 0.9067 | 0.9098 | 0.9080 | 0.7762 | 0.7920 | 0.7835 |
| PBJE† | 0.7922 | 0.7384 | 0.7637 | 0.9077 | 0.8691 | 0.8876 | 0.8179 | 0.7609 | 0.7878 |
| ECPE-MTL† | 0.7548 | 0.7557 | 0.7503 | 0.9093 | 0.8922 | 0.9004 | 0.7769 | 0.7739 | 0.7749 |
| MMN† | 0.7611 | 0.7396 | 0.7502 | 0.9037 | 0.8785 | 0.8907 | 0.7901 | 0.7554 | 0.7721 |
| EPO-ECPE† | 0.7900 | 0.6021 | 0.6824 | 0.9780 | 0.7848 | 0.8702 | 0.7961 | 0.6039 | 0.6848 |
| GAT-ECPE | 0.7265 | 0.7752 | 0.7492 | 0.9098 | 0.9103 | 0.9099 | 0.7617 | 0.7872 | 0.7734 |
| TransECPE† | 0.7708 | 0.6532 | 0.7072 | 0.8879 | 0.8315 | 0.8588 | 0.7874 | 0.6689 | 0.7233 |
| UTOS† | 0.7389 | 0.7062 | 0.7203 | 0.8815 | 0.8321 | 0.8556 | 0.7671 | 0.7320 | 0.7471 |
| Refinement† | 0.7746 | 0.7199 | 0.7463 | 0.8711 | 0.8178 | 0.8436 | 0.7947 | 0.7404 | 0.7666 |
| Guided-QA† | 0.7710 | 0.6920 | 0.7290 | 0.8470 | 0.9080 | 0.8760 | 0.7190 | 0.7920 | 0.7540 |
| MM-R† | 0.8218 | 0.7927 | 0.8062 | 0.9738 | 0.9038 | 0.9370 | 0.8328 | 0.7964 | 0.8135 |
| CD-MRC† | 0.8333 | 0.7800 | 0.8013 | 0.9692 | 0.9380 | 0.9537 | 0.8101 | 0.8068 | 0.8077 |
| CFC-ECPE† | 0.8249 | 0.8125 | 0.8187 | 0.9708 | 0.9332 | 0.9512 | 0.8409 | 0.8116 | 0.8247 |
| RL-TSM | 0.7604 | 0.7584 | 0.7590 | 0.8843 | 0.8334 | 0.8564 | 0.7965 | 0.7739 | 0.7848 |
| MV-SHIF | 0.8500 | 0.8070 | 0.8280 | 0.9670 | 0.9070 | 0.9360 | 0.8410 | 0.7940 | 0.8170 |
| EoCP | 0.7920 | 0.7694 | 0.7842 | 0.9796 | 0.8523 | 0.9113 | 0.8240 | 0.7418 | 0.7921 |
| UECA-Prompt | 0.7182 | 0.7799 | 0.7470 | 0.8475 | 0.9195 | 0.8816 | 0.7624 | 0.7916 | 0.7755 |
| GPT3.5 | 0.4074 | 0.6754 | 0.5082 | - | - | - | - | - | - |
| GPT3.5 DECC | 0.6123 | 0.8156 | 0.6995 | - | - | - | - | - | - |
| **SSG-ECPE** | **0.9947** | **0.9924** | **0.9936** | **0.9995** | **0.9972** | **0.9983** | **0.9952** | **0.9929** | **0.9940** |
| SSG-ECPE-(w/o CPA) | 0.5571 | 0.5545 | 0.5559 | 0.7952 | 0.7915 | 0.7932 | 0.6190 | 0.6161 | 0.6175 |
| **Setting 2: 80% for training, 10% for validation, 10% for testing** | | | | | | | | | |
| Tagging† | 0.7243 | 0.6366 | 0.6776 | 0.8196 | 0.7329 | 0.7739 | 0.7490 | 0.6602 | 0.7018 |
| TransECPE† | 0.7374 | 0.6307 | 0.6799 | 0.8716 | 0.8244 | 0.8474 | 0.7562 | 0.6471 | 0.6974 |
| UTOS† | 0.7104 | 0.6812 | 0.6907 | 0.8649 | 0.8293 | 0.8491 | 0.7418 | 0.7084 | 0.7281 |
| RANKCP† | 0.6575 | 0.7305 | 0.6915 | 0.8936 | 0.8948 | 0.8942 | 0.6940 | 0.7471 | 0.7191 |
| Refinement† | 0.7377 | 0.6802 | 0.7078 | 0.8593 | 0.7993 | 0.8282 | 0.7614 | 0.7039 | 0.7315 |
| PairGCN† | 0.7672 | 0.6791 | 0.7202 | 0.8857 | 0.7958 | 0.8375 | 0.7907 | 0.6928 | 0.7375 |
| ECPE-MLL† | 0.7488 | 0.6976 | 0.7220 | 0.8465 | 0.8990 | 0.8717 | 0.7051 | 0.7704 | 0.7358 |
| MM-R† | 0.7897 | 0.7532 | 0.7706 | 0.9609 | 0.8809 | 0.9188 | 0.8090 | 0.7621 | 0.7845 |
| CD-MRC† | 0.7739 | 0.7478 | 0.7598 | 0.9592 | 0.9183 | 0.9381 | 0.7789 | 0.7616 | 0.7694 |
| **SSG-ECPE** | **0.9964** | **0.9955** | **0.9960** | **0.9995** | **0.9986** | **0.9991** | **0.9967** | **0.9957** | **0.9962** |
| SSG-ECPE-(w/o CPA) | 0.4899 | 0.4894 | 0.4897 | 0.7585 | 0.7578 | 0.7581 | 0.5601 | 0.5596 | 0.5598 |

Table 2: The performance of SSG-ECPE with other benchmark methods on the Chinese dataset for the ECPE task as well as the two auxiliary tasks: EE and CE. The approach with † means using BERT as the pre-trained model.

## 4.2 MAIN RESULTS AND DISCUSSION

**A. Results on Chinese Dataset.** Table 2 shows the performance of SSG-ECPE on the Chinese dataset under two standard splits. Our method achieves SOTA results, with F1 scores exceeding 99% on EE, CE, and ECPE tasks. The key to this performance is the CPA mechanism. As shown in Table 4, removing CPA causes a drastic drop of approximately 50% in F1, revealing severe issues in the base generative model, such as hallucinating clauses or producing lexically inconsistent variants. CPA mitigates these errors by aligning each predicted clause to the most similar one in the input text, ensuring output fidelity. To validate the robustness of CPA, we conduct a threshold sensitivity analysis (see Table 5), showing that F1 peaks around threshold=0.5 and degrades at higher or lower values. This confirms that the high performance is not due to over-permissive matching, but stems from a well-calibrated correction mechanism. While the generative multi-task framework enables expressive joint modeling, CPA stabilizes decoding and transforms SSG-ECPE into a reliable extractor. We randomly sampled 100 test instances and manually verified the predictions. Over 98% of the predicted emotion-cause pairs exactly matched the ground truth, with most errors occurring in cases involving implicit causes or ambiguous clause boundaries. Additional robustness checks, including manual verification, comparisons different pre-trained models, and case studies, are presented in Appendix E, G, and F.

**B. Results on English Dataset.** SSG-ECPE achieves 71.54 F1 on the English dataset, setting a new SOTA (+10.8 over best baseline). The performance gap relative to the Chinese dataset ( 99%) stems from higher ambiguity in novel-derived English texts (implicit causes, subjective annotations) and

| Method | Emotion-Cause Pair Extraction | | | Emotion Clause Extraction | | | Cause Clause Extraction | | |
|---|---|---|---|---|---|---|---|---|---|
| | P | R | F1 | P | R | F1 | P | R | F1 |
| Indep | 0.4694 | 0.4102 | 0.4367 | 0.6741 | 0.7160 | 0.6940 | 0.6039 | 0.4734 | 0.5301 |
| ECPE-2D | 0.6049 | 0.4384 | 0.5073 | 0.7435 | 0.6968 | 0.7189 | 0.6491 | 0.5353 | 0.5855 |
| ECPE-MLL | 0.5926 | 0.4530 | 0.5121 | 0.7546 | 0.6996 | 0.7255 | 0.6350 | 0.5919 | 0.6110 |
| E2E-PExtE | 0.5134 | 0.4929 | 0.5017 | 0.7163 | 0.6749 | 0.6943 | 0.6636 | 0.4375 | 0.5226 |
| IA-ECPE | 0.6014 | 0.4303 | 0.5005 | 0.7398 | 0.6985 | 0.7180 | 0.6387 | 0.5455 | 0.5880 |
| GPT3.5 | 0.4211 | 0.3934 | 0.4068 | - | - | - | - | - | - |
| GPT3.5 DECC | 0.4689 | 0.5442 | 0.5035 | - | - | - | - | - | - |
| SSG-ECPE | **0.7159** | **0.7149** | **0.7154** | **0.7843** | **0.7832** | **0.7837** | **0.7480** | **0.7469** | **0.7474** |

Table 3: Performance comparison on the English ECPE benchmark.

greater linguistic complexity. Nonetheless, consistent gains confirm strong cross-lingual generalization.

| Approach | Emotion-Cause Pair Extraction | | | Emotion Clause Extraction | | |
|---|---|---|---|---|---|---|
| | P | R | F1 | P | R | F1 |
| SSG-ECPE | 0.9947 | 0.9924 | 0.9936 | 0.9991 | 0.9988 | 0.9990 |
| **-w/o CPA** | **0.5016** | **0.5004** | **0.5010** | 0.9823 | 0.9890 | 0.9856 |
| -w/o EE | 0.9956 | 0.9947 | 0.9952 | - | - | - |
| -w/o semantic labels | 0.9948 | 0.9952 | 0.9950 | 0.8600 | 0.8236 | 0.8414 |
| -w/o clause types | 0.9899 | 0.9915 | 0.9907 | 0.9985 | 0.9990 | 0.9987 |
| -w/o emotion types | 0.9925 | 0.9882 | 0.9903 | 0.9963 | 0.9999 | 0.9982 |
| -w/o trigger words | 0.9952 | 0.9858 | 0.9905 | 0.9911 | 0.9961 | 0.9936 |
| -w/o clause ID | 0.9861 | 0.9832 | 0.9847 | 0.9984 | 0.9994 | 0.9989 |
| -w/o task prefix | 0.9962 | 0.9938 | 0.9950 | 0.6591 | 0.9778 | 0.8784 |

Table 4: Ablation study results. Removing CPA leads to catastrophic performance drop.

**C. Ablation Study on Key Components.** Table 4 shows key ablations. Removing CPA collapses ECPE-F1 by ∼50%, confirming its essential role in grounding generation. Omitting task prefixes harms EE (F1: 87.84%), while clause IDs have little effect. Semantic labels (types, triggers) contribute moderately. Interestingly, excluding EE slightly boosts ECPE (+0.16%), but full multi-tasking ensures balanced performance.

**D. Threshold Sensitivity of CPA.** We vary the threshold $\theta_{emo}$ from 0.3 to 0.9. As shown in Table 5, ECPE-F1 fluctuates within a narrow range ($< 1.3\%$), peaking at $\theta_{emo} = 0.5$. Even under extreme settings (e.g., $\theta_{emo} = 0.9$), performance remains high (96.9% F1), far exceeding the best baseline ($\leq 71\%$). This demonstrates that CPA's gains are robust and not due to fine-tuned thresholds. Combined with ablation and cross-lingual results, this confirms the stability and general effectiveness of our approach.

| Threshold | EE-F1 | CE-F1 | ECPE-F1 |
|---|---|---|---|
| 0.3 | 98.21 | 97.45 | 97.98 |
| 0.4 | 99.01 | 98.76 | 99.15 |
| **0.5** | **99.83** | **99.40** | **99.36** |
| 0.6 | 99.72 | 99.31 | 99.10 |
| 0.7 | 99.54 | 99.02 | 98.89 |
| 0.8 | 99.20 | 98.47 | 98.35 |
| 0.9 | 97.98 | 96.65 | 96.92 |

Table 5: F1 scores under different CPA thresholds. Best results at threshold=0.5.

## 5 CONCLUSION

We proposed SSG-ECPE, a semantics-structured generation framework enhanced by clause prediction alignment (CPA) for Emotion-Cause Pair Extraction. By reformulating ECPE as a multi-task generation problem, our model integrates label semantics (e.g., emotion types, roles) into structured outputs and grounds predictions in the input text. Extensive experiments demonstrate that CPA is not only crucial for correcting hallucinations but also substantially improves stability across datasets and thresholds, leading to state-of-the-art results. This highlights a broader insight: lightweight alignment mechanisms can bridge the gap between discriminative and generative paradigms, enabling generative models to achieve both flexibility and reliability in structured extraction tasks. Future work may extend this paradigm to other relation extraction and event argument extraction problems, where grounding generation in the source text is equally vital.

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

# A USE OF LARGE LANGUAGE MODELS (LLMS)

In preparing this paper, we used Large Language Models (LLMs) solely as an assistive tool for improving the clarity and readability of the manuscript. Specifically, LLMs were employed for language polishing, such as grammar correction, style refinement, and phrasing suggestions.

No part of the research process, including problem formulation, dataset construction, methodology design, experimental execution, or result analysis, was conducted by or delegated to LLMs. All scientific contributions, including conceptualization, implementation, and interpretation, are the authors' original work. The authors take full responsibility for the content of this paper.

# B IMPLEMENTATION OF CLAUSE PREDICTION ALIGNMENT (CPA)

We provide a pseudo-code description of the Clause Prediction Alignment (CPA) module used during inference. This implementation directly corresponds to the description in Section 3.5 and is based on Python's `difflib.SequenceMatcher`. The CPA module can be implemented using the Python standard library `difflib`. No external dependencies are required beyond standard Python, ensuring reproducibility and lightweight integration into the inference pipeline.

---
**Algorithm 1:** Clause Prediction Alignment (CPA)

---
**Input:** Predicted clause $c$, set of original document clauses $D$, similarity threshold $\tau$
**Output:** Aligned clause $\hat{c}$

$\hat{c} \leftarrow c$ // Default: retain predicted clause
$best\_sim \leftarrow 0$ ;
**foreach** $c' \in D$ **do**
  $sim \leftarrow \texttt{SequenceMatcher}(c, c').ratio()$ ;
  **if** $sim > best\_sim$ **then**
    $best\_sim \leftarrow sim$ ;
    $\hat{c} \leftarrow c'$ ;

**if** $best\_sim < \tau$ **then**
  $\hat{c} \leftarrow c$ // No reliable match, revert to original
**return** $\hat{c}$

---

**Complexity Analysis.** Let $|D|$ denote the number of clauses in the document and $L$ the average length (in tokens/characters) of a clause. The `SequenceMatcher` function computes the edit similarity between two strings in $O(L^2)$ time. Since CPA compares the predicted clause against all clauses in $D$, the overall time complexity is: $\mathcal{O}(|D| \cdot L^2)$. In practice, $|D|$ is typically small (dozens of clauses per document), and $L$ is short (one or two sentences). Thus, CPA adds negligible computational overhead relative to model inference while significantly improving alignment robustness.

# C EXPERIMENT SETTINGS

To ensure a fair and reproducible comparison with existing approaches, we adopt standard evaluation metrics, including Precision (P), Recall (R), and F1-score, defined as follows:

$$P = \frac{\sum correct\_pairs}{\sum proposed\_pairs} \tag{12}$$

$$R = \frac{\sum correct\_pairs}{\sum actual\_pairs} \tag{13}$$

$$F1 = \frac{2 \times P \times R}{P + R} \tag{14}$$

where $proposed\_pairs$ means the pairs generated by the model, $actual\_pairs$ is the number of actual pairs labeled in the dataset, and the $correct\_pairs$ represents the number of pairs correctly predicted.

For the Chinese dataset, we consider two widely used data splits: 9:1 (train:test) Xia & Ding (2019) and 8:1:1 (train:valid:test) Fan et al. (2020). The pre-trained model used for Chinese experiments is *Randeng-T5-77M-MultiTask-Chinese*. For the English dataset, we follow the 8:1:1 Singh et al. (2021) splitting strategy, using *T5-small* as the backbone.

We employ the AdamW optimizer with an initial learning rate of $3 \times 10^{-4}$ and weight decay of $1 \times 10^{-2}$. The batch size is set to 24, and the model is trained for 20 epochs on all datasets. We apply linear learning rate decay with a warm-up ratio of 0.1 and use gradient clipping at a maximum norm of 1.0 to stabilize training. For Setting 2 and the English dataset, the best model is selected based on validation performance, and early stopping is applied with a patience of 5 epochs. Each experiment is repeated 5 times with different random seeds, and the average performance is reported.

All experiments are conducted on a single NVIDIA GeForce RTX 3090 GPU with 24GB VRAM. A complete 10-fold cross-validation on the Chinese dataset (with a 9:1 train:test split in each fold) takes approximately 27 hours, which is comparable to the time required for BERT fine-tuning ( 30 hours).

## D  DATASET STATISTICS

Table 6 summarizes the statistics of the two benchmark datasets. Most documents contain only a single emotion-cause pair (ECP), accounting for 89.77% in the Chinese dataset and 89.24% in the English dataset, which indicates that the majority of instances follow relatively simple document structures.

On average, Chinese documents are longer (14.77 clauses per document) compared to English ones (7.67 clauses per document). In both datasets, most ECPs occur within a short clause distance: 95.8% in Chinese and 91.73% in English appear within three clauses. This reflects a strong local co-occurrence tendency, which models can easily exploit but also creates a positional bias.

Such bias may hinder the detection of complex, long-distance emotion–cause relationships, particularly in the Chinese dataset where longer document lengths (up to 73 clauses) and wider clause distances (up to 12) necessitate capturing semantic dependencies beyond local proximity. These observations highlight two major challenges in ECPE: (1) mitigating positional bias and (2) modeling semantic coherence across long documents.

|  | Chinese Dataset | | English Dataset | |
| --- | --- | --- | --- | --- |
| Range | Num. | Ratio | Num. | Ratio |
| Documents | 1945 | 100% | 2843 | 100% |
| 1 ECP | 1746 | 89.77% | 2537 | 89.24% |
| 2 ECPs | 177 | 9.10% | 256 | 9.00% |
| >2 ECPs | 22 | 1.13% | 50 | 1.76% |
| Abs dist. = 0 ECPs | 511 | 23.6% | 1640 | 51.01% |
| Abs dist. = 1 ECPs | 1342 | 61.9% | 825 | 25.66% |
| Abs dist. = 2 ECPs | 224 | 10.3% | 328 | 10.21% |
| Abs dist. = 3 ECPs | 50 | 2.3% | 156 | 4.85% |
| Abs dist. >3 ECPs | 40 | 1.9% | 266 | 8.27% |
| ECPs | 2154 | - | 3215 | - |
| Emotion clause | 2085 | - | 2872 | - |
| Cause clause | 2142 | - | 3187 | - |
| Avg clauses/doc | 14.77 | - | 7.67 | - |
| Max clauses/doc | 73 | - | 41 | - |
| Avg ECPs/doc | 1.11 | - | 1.13 | - |
| Max ECPs/doc | 4 | - | 6 | - |
| Min dist. emo&cau | 0 | - | 0 | - |
| Max dist. emo&cau | 12 | - | 25 | - |
| **Avg dist. emo–cau** | **0.94** | - | **1.54** | - |

Table 6: Dataset statistics for the two benchmark dataset used for ECPE task.

## E  MANUALLY VERIFIED THE CPA'S PREDICTIONS

To further validate the reliability of our Clause Prediction Alignment (CPA) mechanism, we conducted a manual verification of predicted emotion-cause pairs on a random subset of the test set.

**Verification Procedure.** We randomly sampled 50 instances from the Chinese and English test set. For each instance, human annotators checked whether each predicted emotion-cause pair exactly

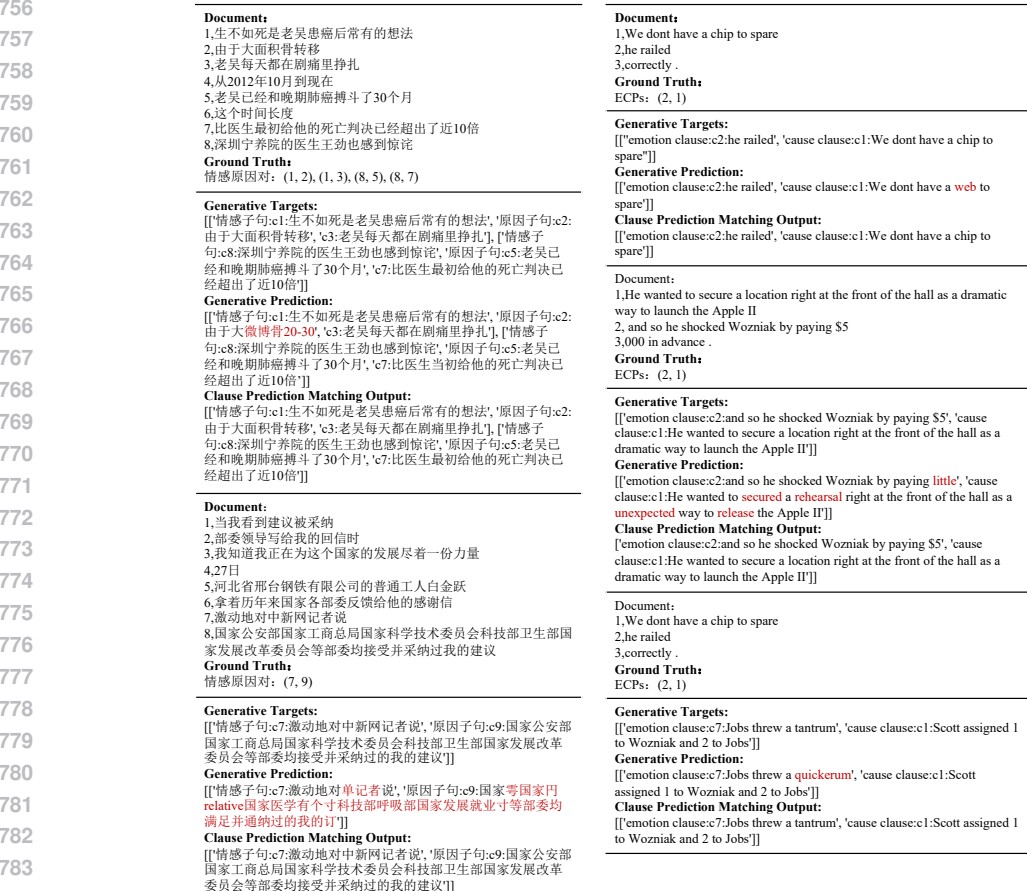

Figure 3: Case studies.

matched the ground truth, including correct clause boundaries and roles. Predictions were categorized as: **Exact Match**: Both emotion and cause clauses match the ground truth. **Partial Match**: Either the emotion or cause clause is partially correct. **Mismatch**: Both clauses are incorrect.

On the Chinese dataset, 48 out of 50 instances achieved *Exact Match*, yielding a 96% exact accuracy. The two non-exact cases involved: A lexical deviation ("Teresa" vs. "Graduates") in the raw generation, which was corrected by CPA to the correct clause via similarity matching. An ambiguous multi-pair case where one cause clause was missed due to overlapping semantics. On the English dataset, 49 out of 50 instances were exact matches. The single error occurred in a narrative with an implicit causal relation ("he felt guilty because he didn't help"), where the cause is not explicitly stated. Notably, when inspecting model outputs *without CPA*, we observed frequent hallucinations—generating clauses not present in the original text. In contrast, CPA ensures all predictions are grounded in actual input clauses, effectively eliminating spurious generations. These results confirm that the high F1 scores reflect genuine prediction accuracy and demonstrate that CPA plays a critical role in aligning generated outputs with the input text.

# F  CASE STUDIES ON CHINESE AND ENGLISH DATASETS

To illustrate the behavior of SSG-ECPE and the role of Clause Prediction Alignment (CPA), we present one representative example from each language.

On the Chinese dataset, the raw generator produces significant errors: ("...20-30") is a hallucinated phrase not in the original text, likely due to noisy token generation. Additionally, ("originally") slightly deviates from ("initially"). Without CPA, these would be false predictions. However, CPA

aligns each predicted clause with the most similar one in the document, successfully recovering the correct clauses and producing perfect matches. On the English dataset, the generator incorrectly substitutes "chip" with "web", a semantically plausible but lexically incorrect word. This type of error, common in autoregressive models, would lead to an exact-match failure. CPA detects that the similarity between "We dont have a web to spare" and the true clause "We dont have a chip to spare" is high (e.g., SequenceMatcher ratio $> 0.9$), and maps it back to the closest valid clause in the input. Thus, the final output becomes accurate despite imperfect generation.

## G PERFORMANCE COMPARISON OF PRE-TRAINED MODELS ON ECPE TASK

To analyze the performance difference between different pre-trained models on the Emotion Cause Pair Extraction (ECPE) task, we performed comparative experiments on the Randeng-T5 model used for the Chinese dataset. Due to the limitation of computational resources, we selected mBART-large-50[1] and mT5-small[2] as comparison models.

BART employs a denoising autoencoder architecture, with a bidirectional encoder and a causal decoder, which differs from the unified text-to-text framework of T5-based models. Both `mT5` and `Randeng-T5` are based on the T5 architecture but differ in their pre-training corpora. `mT5` is trained on the multilingual corpus, which includes a small proportion of Chinese text among 101 languages. In contrast, `Randeng-T5` is specifically optimized for Chinese, with its pre-training corpus exclusively composed of Chinese text.

On the Chinese dataset, `Randeng-T5-77M` and `mT5-small` achieve strong performance, benefiting from their exposure to Chinese text during pre-training. Notably, `Randeng-T5-77M`, despite its smaller size (77M parameters), matches the performance of the larger `mT5-small` (300M), underscoring the critical importance of language-specific pre-training for optimal performance.

On the English dataset, `T5-base` achieves the best results, demonstrating its effectiveness in generation-based information extraction. `mT5-small` performs slightly worse, likely due to interference from non-English content in its multilingual training data. Once again, `BART` underperforms dramatically (F1: 2.38%), with nearly all outputs failing to conform to the expected structure. This consistent failure across languages highlights a key insight: for structured generation tasks like ECPE, the choice of pre-training objective (e.g., denoising vs. span corruption) and language specialization critically affects model compatibility and performance.

Moreover, BART performed significantly worse, which can be attributed to the following factors: 1. Limited training objective: BART is pre-trained on unsupervised data with denoising reconstruction as its primary task, resulting in a narrow focus that limits its generalization ability for the ECPE task. 2. Mismatch with task requirements: The pre-training objective of BART is mainly to restore corrupted input, which, while beneficial for generation tasks, may fall short in capturing the complex causal relationships required for ECPE. 3. Architectural limitations: BART's design is better suited for tasks like text summarization but is less effective for tasks demanding more substantial reasoning and semantic understanding. Additionally, considering the model size, mBART-large-50 has 610M parameters, Randeng-T5-77M has 77M parameters, and mT5-small has 300M parameters. Although Randeng-T5 is relatively lightweight in scale, it performs strongly on the Chinese dataset. Its advantage is due to its pre-training optimization explicitly tailored for Chinese corpora, allowing it to capture better linguistic patterns and discourse structures common in Chinese writing.

| Dataset | PLM | P | R | F1 |
|---|---|---|---|---|
| | Randeng-T5 | 0.9947 | 0.9924 | 0.9936 |
| Chinese | mT5 | 0.9952 | 0.9947 | 0.9949 |
| | BART | 0.4851 | 0.4692 | 0.4767 |
| | T5 | 0.7159 | 0.7149 | 0.7154 |
| English | mT5 | 0.7082 | 0.6900 | 0.7013 |
| | BART | 0.0242 | 0.0237 | 0.0238 |

Table 7: Different Pre-trained Language Models (PLM) perform on ECPE task. We report results for two PLMs, Rangdeng, mT5, and BART on Chinese and English dataset.

---

[1]https://huggingface.co/facebook/mbart-large-50

[2]https://huggingface.co/google/mt5-small

## H  SUPPLEMENTARY RESULTS FOR LLM-BASED APPROACHES

To contextualize our method within the landscape of large language models (LLMs), we conduct preliminary evaluations of several LLM-based approaches, including GPT-4o, DeepSeek-R1, and DeepSeek-V3, in addition to GPT-3.5. Due to the high computational cost (GPT-4o:$500) and API expenses of LLMs, comprehensive evaluations are resource-intensive. Following the setup of Wang et al. (2023b), we perform zero-shot evaluations on a subset of 100 test samples from the Chinese dataset. As shown in Table 8, GPT-4o, DeepSeek-V3, and DeepSeek-R1 achieve F1 scores of 55.82%, 63.72%, and 64.37%, respectively.

Our SSG-ECPE model, evaluated on the same 100-sample subset, achieves a significantly higher F1 score of 99.36%. This substantial performance gap highlights the effectiveness of our semantics-structured generation and clause-level alignment framework. The results suggest that even state-of-the-art LLMs face challenges in ECPE under zero-shot conditions, potentially due to the need for precise clause-level matching and the complexity of identifying multiple, potentially nested emotion-cause pairs.

| Method | P | R | F1 |
|---|---|---|---|
| GPT3.5 (prompt, 0-shot) | 54.13 | 50.86 | 52.44 |
| GPT3.5 (DECC, 0-shot) | 57.50 | 39.66 | 46.94 |
| GPT-4o | 58.64 | 53.26 | 55.82 |
| DeepSeek-V3(0-shot) | 65.45 | 62.07 | 63.72 |
| DeepSeek-R1(0-shot) | 65.97 | 62.86 | 64.37 |
| SSG-ECPE | 99.45 | 99.28 | 99.36 |

Table 8: Results of SSG-ECPE with other LLMs-based baselines on the Chinese dataset for the ECPE task.

## I  COMPARISON ON EXTRACTING MULTIPLE PAIRS.

The extraction of multiple emotion-cause pairs (ECPs) within a single document is a significant challenge due to potential nesting, overlap, and complex inter-clause dependencies. To evaluate the robustness of our method in such complex scenarios, we partitioned the test set (from the 10-fold CV) of the Chinese dataset into two subsets: one containing documents with only one ECP, and the other containing documents with two or more ECPs.

Table 9 presents the results. SSG-ECPE achieves state-of-the-art performance on both subsets. Notably, on documents with multiple ECPs, SSG-ECPE significantly outperforms all baselines, achieving an F1 score of 98.70%, which is over 38 points higher than the best competing method (EPO-ECPE, 60.19%). This dramatic performance gap underscores the effectiveness of our semantics-structured generation and clause-level alignment framework in handling complex, multi-pair documents.

| Approach | P | R | F1 |
|---|---|---|---|
| Single ECP | | | |
| Inter-EC | 0.6734 | 0.5939 | 0.6288 |
| RANKCP | 0.6625 | 0.6966 | 0.6780 |
| ECPE-MLL | 0.6870 | 0.6832 | 0.6851 |
| UTOS | 0.6765 | 0.6232 | 0.6480 |
| EPO-ECPE | 0.7668 | 0.6559 | 0.7065 |
| **SSG-ECPE** | **0.9937** | **0.9900** | **0.9918** |
| Multiple ECPs | | | |
| Inter-EC | 0.5912 | 0.3302 | 0.4206 |
| RANKCP | 0.7508 | 0.4390 | 0.5531 |
| ECPE-MLL | 0.7045 | 0.4776 | 0.5688 |
| UTOS | 0.5545 | 0.4676 | 0.5035 |
| EPO-ECPE | 0.8396 | 0.4768 | 0.6019 |
| **SSG-ECPE** | **0.9822** | **0.9918** | **0.9870** |

Table 9: The results for documents with only one and more than one ECP on the Chinese benchmark dataset.

