# OpenReview forum: "SSG-ECPE: Semantics-Structured Generation with Alignment for Emotion-Cause Pair Extraction"
_ICLR.cc/2026/Conference — ICLR 2026 Conference Withdrawn Submission_

### Official Review · Reviewer_2s1v · 2025-10-15

**Soundness:** 1
**Presentation:** 3
**Contribution:** 1
**Rating:** 2
**Confidence:** 5

**Summary:**

This paper presents SSG-ECPE, a generative multi-task framework for Emotion-Cause Pair Extraction (ECPE) that rethinks the task as a structured text-to-text generation problem. The generation format includes EE generation and ECPE generation. To ensure alignment and clause-level fidelity, the method introduces a Clause Prediction Alignment (CPA) mechanism that grounds generated outputs to input clauses. Experiments on both Chinese and English benchmarks demonstrate the effectiveness.

**Strengths:**

1. Paper is well-written.

2. Strong empirical results: Achieves SOTA performance on both Chinese and English benchmarks with large margins; extensive ablation validates each component.

**Weaknesses:**

1. Limited novelty: applying a generative framework to a non-generative task is not very innovative.
2. I still cannot understand why the proposed method can achieve such remarkably high performance. Could you provide some cases to illustrate when the proposed method might fail?
3. The proposed method additionally relies on emotion types and trigger words, which may cause difficulties in real-world deployment.
4. Applying the alignment mechanism to other NLP tasks could help improve the generalizability of this work.
5. How does the method perform on LLMs such as LLaMA or Qwen?
6. Subjectively, the research in this paper may feel somewhat outdated in 2025.

**Questions:**

Since the authors have already used task-specific prefixes, the two decoders may be unnecessary, as their functions might overlap.

---

### Official Review · Reviewer_3kyw · 2025-10-20

**Soundness:** 2
**Presentation:** 3
**Contribution:** 2
**Rating:** 2
**Confidence:** 4

**Summary:**

This paper presents SSG-ECPE, a generative framework for Emotion–Cause Pair Extraction (ECPE) that rethinks the task as a structured text-to-text generation problem. Unlike traditional discriminative approaches that classify clause pairs or assign labels independently, the proposed model aims to capture richer semantic dependencies across clauses.

The framework introduces a semantics-structured generation strategy, where outputs explicitly encode clause roles, emotion types, and trigger words. Additionally, a Clause Prediction Alignment (CPA) module is designed to align generated clauses with those in the input text, effectively reducing hallucinations and ensuring faithful generation.

Experiments on both Chinese and English ECPE benchmarks show that SSG-ECPE achieves state-of-the-art results, significantly outperforming previous methods. The ablation studies further highlight the importance of CPA, as removing it causes a drastic drop in performance, demonstrating its essential role in maintaining output consistency and reliability.

**Strengths:**

1. The paper proposes a multi-decoder T5-based generative framework that replaces traditional discriminative approaches for ECPE. This design effectively integrates emotion and cause extraction within a unified model and demonstrates strong, consistent performance across multiple datasets and baselines.

2. The proposed **CPA** module is simple yet effective, requiring no additional training while significantly improving generation faithfulness. Its practicality and interpretability make it an insightful contribution for reducing hallucination issues in generative information extraction models.

**Weaknesses:**

1. The proposed method is primarily engineering-oriented, relying on structured output formatting and post-processing rather than introducing substantial methodological or theoretical innovations. While the semantics-structured generation idea is well-executed, it builds heavily on existing T5 architectures and multi-task setups, offering limited novelty beyond task-specific adaptation.

2. The model shows strong dependence on the CPA module. As reported in Table 4, removing CPA leads to an F1 drop of around 50%, indicating that the underlying generative model struggles with clause-level accuracy and relies heavily on post-hoc correction for stability. This suggests that the core generation mechanism lacks robustness and semantic grounding without external alignment.

3. The generalization and reliability of the approach remain uncertain.
    * The datasets used are mainly derived from news and literary texts, leaving performance in real-world or domain-diverse scenarios unclear.
    * The near-perfect results on the Chinese dataset (over 99% F1) raise potential concerns of data leakage or dataset bias. Furthermore, the use of LCS-based similarity instead of exact matching in CPA may inflate reported scores.
    * The alignment thresholds and handcrafted rules make the approach less adaptable to open-domain or noisy free-text environments.

**Questions:**

1. In Figure 1, the authors mention using “various shades of color” to distinguish different emotion and cause clauses, but the visual differences are not easily discernible. Could the figure be improved for clearer visualization (e.g., distinct color schemes or annotations)?


2. (Lines 126–140): The paper claims that prior joint learning frameworks face issues of task imbalance and poor generalization. Could the authors provide empirical evidence or references to support these claims, such as comparative experiments or prior studies demonstrating these limitations?


3. (Section 3.2): The model uses a [TASK] prefix to distinguish between EE and ECPE generation. How does this approach fundamentally differ from conventional multi-task learning paradigms that also use task identifiers or prompts? What specific benefits does this prefix bring beyond standard multi-task setups?


4. (Section 3.5): The introduction (Lines 111–114) states that the CPA mechanism helps reduce hallucinations. However, since CPA only aligns outputs based on textual similarity, could the authors clarify how this similarity-based correction directly mitigates hallucination? Are there quantitative or qualitative analyses (e.g., hallucination rate comparisons) to support this claim?

---

### Official Review · Reviewer_Hivz · 2025-10-24

**Soundness:** 2
**Presentation:** 2
**Contribution:** 2
**Rating:** 2
**Confidence:** 4

**Summary:**

This paper proposes a structured text-generation approach for emotion-cause pair extraction task and uses a similarity-based matching mechanism to align generated outputs with source clauses. The authors evaluate the method on both Chinese and English datasets.

**Strengths:**

1. This paper reformulates the EE/ECPE tasks as semantically structured generation problems and specifies a concrete output protocol and parsing procedure.

2. The proposed model attains strong performance on the Chinese dataset used in the paper, surpassing existing baselines.

3. This paper clearly presents the model architecture and algorithmic implementation details, providing sufficient information to support reproducibility.

**Weaknesses:**

1. **Limited novelty and contribution:** The experimental results suggest that the method’s advantages are largely attributable to the Clause Prediction Alignment (CPA) mechanism. Such alignment via string/semantic similarity is a widely used engineering technique in prior NLP work and is not introduced by this paper. For example:
    - UIE: Lu, Yaojie, et al. “Unified Structure Generation for Universal Information Extraction.” Proceedings of the 60th Annual Meeting of the Association for Computational Linguistics (Volume 1: Long Papers), 2022.
    - Text-to-SQL: Wang, Jun, et al. “Improving Text-to-SQL Semantic Parsing with Fine-grained Query Understanding.” Proceedings of the 2022 Conference on Empirical Methods in Natural Language Processing: Industry Track, 2022.
    - Entity Linking: Loureiro, D., and A. Jorge. “MedLinker: Medical Entity Linking with Neural Representations and Dictionary Matching.” Advances in Information Retrieval: 42nd European Conference on IR Research (ECIR 2020), Proceedings, Part II, Vol. 12036, 2020.

2. **Questionable effectiveness of multi-task training:** Despite the author claims that a shared-encoder/dual-decoder setup promotes cross-task knowledge transfer, Table 4 shows that removing the EE task actually yields higher performance.

3. **Incomplete experimentation:**
    - For several strong baselines (e.g., MV-SHIF, UECA-Prompt), the paper reports results only on the Chinese dataset; their English results which are reported in the original papers are not included in Table 3.
    - The ablation studies are insufficient: removing several factors paradoxically improves the final performance, yet the paper offers no explanation. In addition, ablations on the English dataset and on the Cause Clause Extraction (CE) task are missing.
    - Sensitivity analysis is insufficient: for the key loss-weight parameter $\lambda$, the paper neither reports the specific values used nor provides a sensitivity study.

4. **Many writing/formatting issues and internal inconsistencies:**
    - Threshold symbol mismatch: The CPA threshold is denoted as $\tau$ in the method/algorithm description, but the sensitivity analysis later varies $θ_\text{emo}$.
    - Inconsistent reporting of experimental results: The abstract claims "+21.3 F1 improvement on the English benchmark", whereas the main text states "+10.8 over the best baseline". Table 3 reports "F1=71.54" for SSG-ECPE; the strongest baselines for ECPE is "F1=51.21" (ECPE-MLL), implying gains of "+20.3". These three statements are not aligned.
    - Missing references for baselines: The baselines MV-SHIF and EoCP appear in the main results table, but there are no corresponding entries for them in main text and the references section.
    - Missing citation hyperlinking: The in-text mention "Singh et al. (2021)" is not hyperlinked.
    - Inconsistent between ablation study and main-experiment values: In the ablation study, the results under the "w/o CPA" setting are inconsistent with those reported in Table 2 for the "SSG-ECPE (w/o CPA)" row.

**Questions:**

See the weaknesses above.

---

### Note · Authors · 2025-12-09

I have read and agree with the venue's withdrawal policy on behalf of myself and my co-authors.